# Molecular Modelling Hurdle in the Next-Generation Sequencing Era

**DOI:** 10.3390/ijms23137176

**Published:** 2022-06-28

**Authors:** Guerau Fernandez, Dèlia Yubero, Francesc Palau, Judith Armstrong

**Affiliations:** 1Department of Genetic and Molecular Medicine—IPER, Hospital Sant Joan de Déu, Institut de Recerca Sant Joan de Déu, 08950 Barcelona, Spain; guerau.fernandez@sjd.es (G.F.); francesc.palau@sjd.es (F.P.); judith.armstrong@sjd.es (J.A.); 2Center for Biomedical Research Network on Rare Diseases (CIBERER), ISCIII, 08950 Barcelona, Spain; 3Division of Pediatrics, University of Barcelona School of Medicine and Health Sciences, 08007 Barcelona, Spain

**Keywords:** NGS, VUS, multi-omics, tissue-specific model

## Abstract

There are challenges in the genetic diagnosis of rare diseases, and pursuing an optimal strategy to identify the cause of the disease is one of the main objectives of any clinical genomics unit. A range of techniques are currently used to characterize the genomic variability within the human genome to detect causative variants of specific disorders. With the introduction of next-generation sequencing (NGS) in the clinical setting, geneticists can study single-nucleotide variants (SNVs) throughout the entire exome/genome. In turn, the number of variants to be evaluated per patient has increased significantly, and more information has to be processed and analyzed to determine a proper diagnosis. Roughly 50% of patients with a Mendelian genetic disorder are diagnosed using NGS, but a fair number of patients still suffer a diagnostic odyssey. Due to the inherent diversity of the human population, as more exomes or genomes are sequenced, variants of uncertain significance (VUSs) will increase exponentially. Thus, assigning relevance to a VUS (non-synonymous as well as synonymous) in an undiagnosed patient becomes crucial to assess the proper diagnosis. Multiple algorithms have been used to predict how a specific mutation might affect the protein’s function, but they are far from accurate enough to be conclusive. In this work, we highlight the difficulties of genomic variability determined by NGS that have arisen in diagnosing rare genetic diseases, and how molecular modelling has to be a key component to elucidate the relevance of a specific mutation in the protein’s loss of function or malfunction. We suggest that the creation of a multi-omics data model should improve the classification of pathogenicity for a significant amount of the detected genomic variability. Moreover, we argue how it should be incorporated systematically in the process of variant evaluation to be useful in the clinical setting and the diagnostic pipeline.

## 1. Introduction

Since the release of the first draft of the human genome in 2001 [1], one of the most fundamental challenges clinical geneticists have faced is trying to uncover the cause of Mendelian or single-gene disorders. The lack of functional knowledge of most of the variability within the genome has been the main barrier for genetic diagnostics. In an effort to understand the genetic complexity, rare diseases (RDs) have been the focus of active research. Approximately 72% of RDs are caused by genetic mutations. Due to the singularity of RDs, a disease or condition is considered rare if it affects less than 1 in 200,000 people (United States) or less than 1 in 2000 people (Europe) [2]. Unexplored regions of the genome unlock its function, leading to a distinguishable phenotype. The identification of one or multiple variants that trigger an RD maps a genomic localization to a function (genotype–phenotype). The obvious handicap of working with RDs is the difficulty in establishing a cause–effect association of a specific mutation, requiring functional studies. Although mutation hotspots have been described for specific RDs, most disease-causing genes have pathogenic mutations scattered all over the gene body or regulatory regions. Identifying the same mutation in different patients with the same pathophysiological processes is difficult in the RD context. To solve this conundrum, two strategies have been applied: The first approach involves investigation of familiar or highly consanguineous populations to increase the probability of characterizing recurrent mutations detected throughout a well-described pedigree, which correlates with the phenotype of study. The second procedure requires gathering and aggregating patients with the same phenotype on a global scale. An example of this kind of initiative is Matchmaker Exchange [3], where genotypic and phenotypic data are shared, seeking common variability to reveal the putative etiology of an undiagnosed group of patients. This second procedure provides less sensitive results compared to the first strategy, as genetic alterations among different patients might not be the same. Overall, 9349 clinical entities and 7877 rare-disease-gene linkages have been described [4].

Since 2009, when next-generation sequencing (NGS) was first introduced to identify disease-causing genes [5], the pace of RD-causing gene discovery has increased drastically [6]. Due to its high resolution, being able to identify mutations at the base-pair level, and its capability to analyze multiple genes at the same time, NGS has become one of the most powerful tools to detect genetic variability. Thus, NGS has been fully incorporated into the clinical setting for disease diagnosis, in combination with other, more conventional techniques [7]. The use of NGS has allowed a deep characterization and subgrouping of certain diseases, leading to more accurate diagnosis. However, as a counterbalance to this precise genetic profiling, phenotypic overlap in human diseases has increased since the discovery of new causative genes, blurring the lines between diseases.

NGS has driven genetic diagnosis of RDs to a global 50% of diagnostic yield. Some diseases present diagnostic yields over 70%—for example, inborn errors of metabolism or specific neurological conditions, in which the presence of a biomarker facilitates the genetic diagnosis [8,9]. Moreover, having biological support for a genetic disorder allows the possibility to design other strategies to resolve unsolved conditions, such as looking for intronic or regulatory regions, or using specific approaches to detect structural variation. Unfortunately, the diagnostic odyssey remains harder for patients with RDs that do not fall into these groups, and understanding their genomes becomes a complex process. It is worth mentioning the reusability of NGS data. Variants of uncertain significance (VUSs) detected in an undiagnosed patient can be reclassified as disease-causing in light of new discoveries.

Alongside the significant increase in diagnostic yield, NGS has boosted the amount of data generated and, therefore, has increased uncertainty. The more genomes are sequenced, the more VUSs are obtained. Moreover, due to pleiotropic effects, other factors are putting molecular analysis alone into a deadlock; a single gene could affect multiple and apparently unrelated phenotypes and mutational penetrance, and a particular mutation does not always produce the same effect in all individuals who carry it (Figure 1).

The aim of this work is to portray the actual state of molecular diagnosis, considering only genomic sequencing, and to study complementary sources of information in order to combine them and determine a model to make better use of these powerful data. Omics data have to be structured and integrated in order to achieve a clearer picture of the possible impact that novel and unknown variants might have. It is imperative for the benefit of patients with RDs that we drive diagnostic yield to as close to 100% as possible for as many conditions as possible [10], using all of the tools we have available.

## 2. Results

### 2.1. Genetic Variability

Considering the fraction of DNA sequenced by NGS technology, genomic variation can be determined using gene panels, which can span from less than a hundred to a few thousand protein-coding genes; whole-exome sequencing (WES), which comprises genes with well-established pathogenicity as well as functionally unknown genes; and whole-genome sequencing (WGS), where variants in the intergenic non-coding regions can be analyzed. Panels of all known pathogenic genes—also known as clinical exome sequencing (CES)—are one of the choices to introduce NGS into the clinical setting.

To illustrate the limitations that a regular genetics department encounters routinely when analyzing NGS data, we studied the outcomes derived from the Illumina TruSight One Expanded (TSOex) gene panel (6704 genes) from all patients analyzed over 2 years (n = 2474) at the Sant Joan de Déu Children’s Hospital. Genes included in TSOex are abundant in cellular processes, related to binding or catalytic activity, mostly belong to the metabolite interconversion enzyme protein class, and are enriched in 71 OMIM (Online Mendelian Inheritance in Man) disease categories, with anemia being the most significant (Figure 2A–D and Appendix A).

As genetic variants with low population frequency are more likely to be the cause of genetic diseases, for this study, we selected variants with an allele frequency < 0.01 (European non-Finnish, gnomAD). A total of 2,456,984 variants were detected within all samples. The five most abundant types of variants per sample are shown in Figure 3A; missense mutations were the most prominent. Although synonymous mutations are also very well represented, when determining the unique mutations within all samples, their proportion becomes almost negligible (Figure 3B). The drastic reduction in recurrent synonymous mutations might be the result of common variants in the Southern European population that are underrepresented in the gnomAD database. More than 90% of the mutations are missense, followed by stop-gain and structural interaction mutations. Next, we determined the number of mutations detected per gene, and summarized all samples (Figure 3C). Most genes only showed one mutation. TTN was the most mutated gene, with 2337 individual variants throughout its gene body. Interestingly, there was a slight peak between 8 and 50 mutations per gene, followed by a long tail of increasingly mutated genes. Due to the differential mutational landscape among the analyzed genes, we compared them to their lack of permissiveness to accept missense variations that might lead to loss of protein function, using the observed versus expected values from gnomAD (Figure 3D). The higher (closer to 0) this score, the more tolerant the gene, and the lower this score (negative values), the more intolerant it is. Most genes are tolerant to variations, showing two high-density areas: one at 1 mutation per gene (for example, the case of gene RPS17), and another at close to 50 mutations per gene (for example, the gene LRPAP1). Surprisingly, the most intolerant genes correlate with the 8–50 peak shown in Figure 3C, indicating that mutations in those genes are the most susceptible to leading to loss of function.

### 2.2. Variant Conservation Score

Regions or individual nucleotide positions with low evolutionary variability might indicate negative selection due to functional constraints. The absence of allelic variants and a high conservation index—measured using the CADD (Combined Annotation-Dependent Depletion) index—in such regions could be an indicator of clinical importance when a variant is detected. We classified variants using the SnpEff impact categories (defined in Appendix A), the CADD score, and allele frequency (Figure 4A and Table 1). We consider CADD scores over 20 as having a high probability of being in front of a pathogenic variant. We found the correlation between HIGH (i.e., variants with a disruptive impact on the protein) mutations and CADD scores > 20 (88% within its group). Meanwhile, MODERATE (i.e., variants with a non-disruptive impact on the protein) mutations have 58% high-CADD-score variants, while LOW (i.e., variants unlikely to change protein behavior) and MODIFIER (i.e., variants with no evidence of impact) categories only reach approximately 11.5%. Most mutations have an allele frequency close to zero, indicating that putative deleterious mutations tend to be extremely rare (frequencies < 0.002 are the most common in all categories). It is worth mentioning that a high-concentration area can be observed in the MODERATE mutations, with a low allele frequency and a CADD score just above 20. When analyzing individual samples, there is a linear correlation (r = 0.97) between the total number of variants and the ones with a CADD score > 20 (Figure 4B). Approximately 20% of the mutations detected have a high CADD score independent of the sample’s total mutations. This correlation might be due to the types of genes included in the TSOex gene panel. Thus, we would not expect to observe this same pattern as more regions of the genome are analyzed.

### 2.3. Variant Classification

From the 2474 samples analyzed, approximately 800 (30%) had a single-nucleotide variant (SNV) that could be associated with the patient’s pathophysiological process. According to our own data, diagnostic efficiency using TSOex reaches approximately 50%, considering other genomic variations such as indels or copy-number variants (CNVs). Almost all informed variants (98%) have a CADD score > 20 (Figure 4C), and are categorized as HIGH or MODERATE (43% and 57%, respectively). MODERATE variants have an elevated proportion of VUSs, taking into account the elevated CADD scores (Table 1 and Table 2). Using the American College of Medical Genetics and Genomics (ACMG) guidelines [11], variants can also be classified into five main groups, namely, pathogenic, likely pathogenic, VUS, likely benign, and benign. Taking one sample as an example, variants were assigned to one of these five categories using the VarSome software, which calculates variant impact using the ACMG classification guidelines (Table 2) [12]. Most variants (80%) were categorized as benign or likely benign. A small fraction (<0.4%) were pathogenic or likely pathogenic. A deep phenotypic characterization is crucial to determine whether these pathogenic variants are relevant to a patient’s disease. A thorough study of the overlap between the described pathophysiological alterations of the mutated gene and the patient’s phenotype will lead to a causative informed variant or an unresolved study. Approximately 20% of mutations are VUSs in a randomly selected sample. After removing the benign/likely benign and pathogenic/likely pathogenic variants, a huge amount of uncertainty remains to be analyzed. As expected, most of these VUSs belong to the MODIFIER category, as their relevance in gene function or regulation is not well established. Surprisingly, a significant proportion of VUSs belong to the LOW variant class, highlighting the difficulty in reaching a conclusive diagnosis when taking only genetic data into account.

### 2.4. Expression Variability

Gene expression data are gradually being incorporated into the diagnostic process to detect certain types of variations. Specifically, variability affecting alternative splicing, allele-specific outliers, and expression outliers are of great interest. The relevance of a genomic mutation with weak pathogenic evidence can be increased by its association with an aberrant gene expression profile. Transcriptomic data can be highly relevant, especially when dealing with VUS mutations. Unlike genomic data, expression data are tissue-specific. Thus, genomic variability can be studied by considering the affected target tissue. Most studies trying to address gene expression alterations use blood or fibroblasts as surrogate tissues. The main reason to do so is that these sample types are usually much easier to obtain than the target tissue. Gene expression is extremely diverse among tissues. Clinical diagnosis must exploit transcriptomic data resources such as those generated by international consortia such as GTEx or Cell Atlas. As an example, in Figure 5A we show the expression of the 50 most expressed genes in the brain cortex distributed throughout the 54 tissues collected by the GTEx consortium. These genes are homogeneously expressed in all brain tissues, but differ greatly in others. We want to draw special attention to the difference between whole blood and brain tissues. Thus, when analyzing genomic data and inferring the putative effect of a variant, it is essential to focus on the expression data of the tissue of interest, if available. Moreover, transcript usage might also be crucial when analyzing genetic variability. Most protein-coding genes have more than one transcript (Figure 5B), with 11 being the median. *KCNMA1* has the highest number of transcripts, namely, 92. Not all transcripts are expressed equally in all tissues. Generally, one isoform is predominant, while the rest might be important in a specific tissue or in a specific stage in the cellular/organ genesis.

Transcript switch is not a rare event. From the 54 tissues from GTEx, the switch from the predominant isoform to another in one tissue happens in approximately 1000 genes (Figure 5C). Tung et al. [13] analyzed up to four tissue switches, highlighting the relevance of this phenomenon. *MECP2* has two main transcripts (Figure 6A, Appendix A), and it is shown as an example of a gene with a high-order transcript switch, with 14 tissue switches (Figure 6B). The major cause of Rett syndrome (RTT) is mutations in the *MECP2* gene. RTT is a neurodevelopmental disorder, and transcript switch and differential brain region expression of *MECP2* are crucial to study mutational profiles (Figure 6C). The same *MECP2* isoform is expressed in fibroblasts and the brain, but a different one is expressed in the blood. Integration of genomic and transcriptomic data facilitates variant categorization and prioritization for enhanced diagnosis and clinical decision making.

### 2.5. Protein Variability

Protein abundance is not a replica of gene expression, and deviations at the protein level might not be detectable at the transcript level. Figure 7A shows protein quantification (high, medium, and low) for the same highly expressed genes depicted in Figure 5A (brain cortex). We obtained the data from the Human Protein Atlas. We removed genes for which there was no detectable protein in more than half of the tissues from the plot. There are significant differences between the different cell types within the cortex region, highlighting distinct cellular composition. Protein distribution contrasts with transcript expression. Interestingly, the genes *ACTG1* and *FTH1* have much lower protein levels than their gene expression suggests, while *TUBA1A* and *PSAP* have high protein abundance, as expected.

### 2.6. Protein Structure

Secondary and tertiary protein structure can provide reliable insights into whether a mutation could be associated with disease. Studying protein structure and protein–protein interactions is another source of information to determine whether a mutation might alter the protein’s function. The number of transcripts from the TSOex gene panel with known structures collected from the Protein Data Bank (PDB) is markedly smaller compared with the remaining transcripts to be characterized (Figure 7B). Most genes—approximately 50%—do not have a single transcript with a PDB structure, while only 10% have been fully resolved (Figure 7C).

## 3. Discussion

The incorporation of NGS into clinical diagnostics has uncovered imminent needs in terms of clarifying the huge amount of uncertainty that arises from large-scale sequencing studies. Aside from disease-specific strategies, which allow more detailed and complete designs to reach a molecular diagnosis [14,15,16], generic solutions when confronting any kind of rare disease are difficult to achieve. Predicting the clinical significance of a change at the DNA level is a challenging task due to the myriad of structures/processes that could be affected. In this work, we quantified the number of VUSs in the TSOex gene panel in a single clinical cohort—approximately 20% of the total detected variants, derived from a standard NGS analysis. Although conservation scores are relevant, most pathogenic variants fall into the high CADD scores (Figure 4C), which we showed to be less specific than desired, with a total of 20% of detected variants having high values (Figure 4B). Moreover, we highlighted the importance of taking into account tissue-specific transcriptomic and proteomic data to analyze the phenotypic and clinical outcomes in a more precise manner. The obvious path to elucidate the classification/pathogenicity of a genomic variant is the combination of multiple sources of information. To understand the implications of a mutation in a biological system, it is crucial to grasp the disruption caused in a highly dimensional structure such as a human being [17]. Thus, to capacitate the diagnostic process with all of the available tools and resources, reducing the variant effect uncertainty, we propose a data-aggregated tissue-specific model (Figure 8).

The first step for a targeted strategy as proposed here is accurate phenotyping of the patient’s symptoms and/or congenital malformations. Depending on the suspected disease, the search for the causative variant(s) will focus on a specific tissue or cell type [18]. Highly tissue-specific manifestations of genetic diseases are due to the deregulation of a functional subnetwork of genes (disease module), rather than a single gene [19]. The overall module is responsible for the tissue-specific clinical manifestation. Consequently, distinct etiological disease origins can converge in similar symptoms, leading to phenotypic overlap [20]—a many-to-one relationship. Thus, a precise clinical description may allow clinicians to hypothesize that the cause of the patient’s condition is due to a specific set of genes, narrowing down the genetic analysis to a reduced subset of putative variants. Characterizing the tissue-specific interactome is critical to find the subsets of deregulated modules and their components. Intuitively, the composition of these modules is determined by the gene–phenotype association [21,22,23].

Two main data components are required to build an accurate tissue-specific interactome: gene expression, and protein structure and interactions.

When focusing on a specific disease/tissue, it is important to create a context-specific interactome. The tissue-specific protein interaction landscape must be established to determine the baseline from which a perturbation might lead to the patient’s disease [24,25]. Most protein interaction databases have been compiled from yeast two-hybrid (Y2H) or tandem affinity purification coupled with mass spectrometry (TAP–MS) experiments, without tissue-specific information [26]. To overcome this limitation, transcriptomics has proven to be a crucial tool to prune non-relevant interactions between proteins that in some cases might be associated, but probably not in all scenarios. GTEx and the Human Cell Atlas [27] are two commonly used sources of precise tissue- and cell-type-specific transcriptomic data, respectively, that can be combined with protein interaction data [28]. Co-expression matrices and/or gene regulatory networks can shape the tissue-specific interaction networks [29,30]. Moreover, the combination of gene expression and protein–protein interaction networks might identify gene modules that correlate with disease deregulation. When constructing the different interactomes, isoform differential functionality [31] and expression must be taken into account, as shown in Figure 6. In this regard, another issue arises: depending on the source of information, the coordinates of isoform boundaries might be slightly different. Thus, the nucleotide composition of certain isoforms is questioned. To reduce this uncertainty, a collaborative project called Matched Annotation from NCBI and EMBL-EBI (MANE) has the main objective of reporting a consensus among both reference sets [32]. Still, differences have to be computed when analyzing protein structure and interactions. 

Once the main tissue-specific interactome components have been related, omics data can be combined to associate genomic variability with cell/tissue system disruption. Publicly available data (as mentioned previously) or the patient’s own data can be utilized. To achieve a more accurate analysis, data-intensive precise medicine must be performed, and the patient’s multi-omics data should be collected routinely [33]. Currently, this scenario in daily medical routine is unthinkable, but it should become available as omics technologies become more cost-effective. If transcriptomic and/or proteomic data from the patient are available, even from a surrogate tissue, several complementary analyses can be performed if at least some of the genes contained in the disease module are expressed. From the transcriptomics point of view, aberrant events such as expression and splicing outliers or allelic imbalance can be detected for a more individualized analysis [34,35]. Due to the fact that correlation between mRNA and protein abundance is not always detected, as shown in Figure 5A and Figure 6A, the patient’s transcriptomic data alone might be insufficient to identify the outcome of a genomic mutation. For each tissue there is a balance between both molecules. Thus, disruption of this equilibrium might indicate the source of variability that leads to a pathogenic consequence [36,37].

The presence and abundance of a protein are not the only important factors in deciphering the complexity of the interactome within a tissue; the protein’s three-dimensional (3D) structure/domains and protein–protein interactions are also essential. For decades, structural biologists have studied how to predict the most stable protein folding using methods such as X-ray crystallography, nuclear magnetic resonance spectroscopy (NMR), and cryogenic electron microscopy (cryo-EM). The study of all human proteins using these methods is not possible due to time and cost constraints. Nevertheless, it is imperative to be able to study all proteins capable of triggering a genetic disease, and to study all variants detected by NGS, assigning each of them with potential pathogenicity [38]. Just recently, an algorithm using artificial intelligence, AlphaFold, has shed some light on the prediction of a protein’s correct conformation [39]. The use of multi-sequence analysis provides the possibility to transform a nucleotide sequence into a physiologically stable 3D protein structure. A huge leap forward has been accomplished by the use of deep learning algorithms. This strategy is not only limited to monomeric proteins—the same concept behind AlphaFold is being applied to multimers [40]. Furthermore, in the quest to model tissue-specific interactomes, allosteric sites are an important attribute when studying protein conformation. To map long-range communications between protein regions, a global scan of all proteins must be accomplished to ensure that variability in those sites is accountable for the putative interaction deregulation [41]. 

Building a specialized protein–protein interaction network (PPIN) depends heavily on prior knowledge. In addition to the protein interaction databases that base part of their protein relationship on previously mentioned techniques such as Y2H [42,43,44], predicted interactions can be established using several machine learning techniques. Some methods predict protein interactions using amino acid sequences and support-vector machine (SVM) algorithms with k-nearest neighbor with local description, conventional autocovariance, or deep neural networks with amphiphilic pseudo amino acid composition [45,46,47]. Other approaches use a genomic sequence evolutionary perspective or combination with a learning algorithm [48,49,50]. More recently, due to the fact that two proteins are more likely to interact if they share a common biological process or are present in the same subcellular compartment, semantic similarity has been used to predict PPINs [51]. Moreover, in specific disease scenarios it is fundamental to define the protein’s subcellular localization [52], and combining Gene Ontology with SVM to predict protein interactions might be fundamental to elucidating the proper interactome. New technologies such as PROPER-seq that map protein interactions at a massive scale can help create accurate and specialized PPINs [53]. Importantly, cohesion and separation indices, as well as topological features (i.e., centrality, clustering, or node degrees), are relevant to define interactions between proteins in a PPIN [54].

Although the tissue-specific model presented in this study has been introduced in the context of rare Mendelian disease diagnostics, it is worth mentioning that it could also be applied as a more general model. Two prominent examples are polygenic and late-onset diseases. The former comprise related variants as candidate genes that compose disease modules. The combined effect of multiple mutations within the same subnetwork can be characterized. Polygenic risk scores can also be used to weight the interactions between the components of the interactome. Using this kind of approach, new disease-related genes can be discovered, and the specific relevance of each of their members can be measured [55]. The latter relates to ageing and dynamic evolution through tissue development. While at early stages of life the differentiation process requires most of the organism’s energy, during ageing there is a reverse effect, leading to loss of tissue and cellular identity [56,57]. The interactome’s trajectories can be shaped, and deviations over time can be modelled in a pathophysiological context [58].

Despite recent advancements in the omics realm, several limitations still prevent full characterization of variant pathogenicity. We have presented the difficulties in determining the effects that SNVs can trigger, but other sources of genomic variability might present the same challenges. Nevertheless, mutations at the gene regulatory level (e.g., sncRNAs, promoters, epigenetic signatures, or enhancers), dynamic expansions, or mutations at the genomic level (e.g., structural variants or genomic architecture) can also benefit from an interactome model [59,60]. We believe that with the introduction of new high-throughput technology—particularly in the proteomics area—and integrative algorithms to combine multidimensional data, variant effect uncertainty will be greatly reduced with a tissue-specific model.

## 4. Materials and Methods

### 4.1. Samples

DNA from 2474 pediatric patients with a rare disease was extracted from blood sampled over a 2-year period (June 2018–June 2020) at the Sant Joan de Déu Children’s Hospital. Samples were processed to capture the regions designed on the TruSight One expanded Illumina commercial gene panel (Illumina Inc., San Diego, CA, USA), which includes the coding regions and flanking intronic regions from approximately 6700 genes with known clinical phenotype association, following the protocol instructions and sequenced using a NextSeq 500 instrument (Illumina Inc., San Diego, CA, USA).

### 4.2. Variant Calling

Approximately 6700 genes from the TruSight One expanded clinical exome (Illumina Inc., San Diego, CA, USA) were analyzed per sample. Briefly, FASTQ files were generated using a NextSeq 500 sequencer (Illumina Inc. FastQC v0.11.5 software was used to evaluate read/base quality [61]). Adaptors and low-quality bases (Phred score < 20) were removed using cutadapt software [62]. Reads were aligned to the reference genome HG19 using BWA-MEM [63], and variant calling was performed using GATK 3.7 [64], DeepVariant v0.10.0 [65], and Octopus v0.6.3 [66]. In 807 samples, at least one pathogenic mutation was identified and reported. CNVs were not analyzed in this study.

### 4.3. Annotation

Mutations with gnomAD v 2.1.1 [67] European non-Finnish frequencies > 0.01 were removed from downstream analysis. CADDv1.3 and the observed versus expected ratio from gnomAD were annotated using SnpEff to determine the impacts specific mutations might have on the protein function. Plots were generated using R language with the ggplot package.

### 4.4. Gene/Disease Classification

PANTHER 16.0 was used to group TSOex genes in different categories [68]. OMIM disease enrichment analysis was performed using EnrichR [69]. Clusters were computed using the Leiden algorithm. Disease terms were plotted on the first two UMAP dimensions.

### 4.5. Databases

Transcriptomics data were collected from the GTEx web portal [70] on 12 October 2021. Brain *MECP2* expression (TPMs) from GTEx was visualized using the R package cerebroViz [71]. Transcript switch and quantification were obtained from Top-Ranked Transcript Isoforms in Human Protein-Coding Genes (TREGT) [72]. Transcript PDB IDs were extracted from ENSEMBL BioMart on 12 October 2021 [73]. Protein quantification from specific tissue cell types was retrieved from the Human Protein Atlas [74].

## Figures and Tables

**Figure 1 ijms-23-07176-f001:**
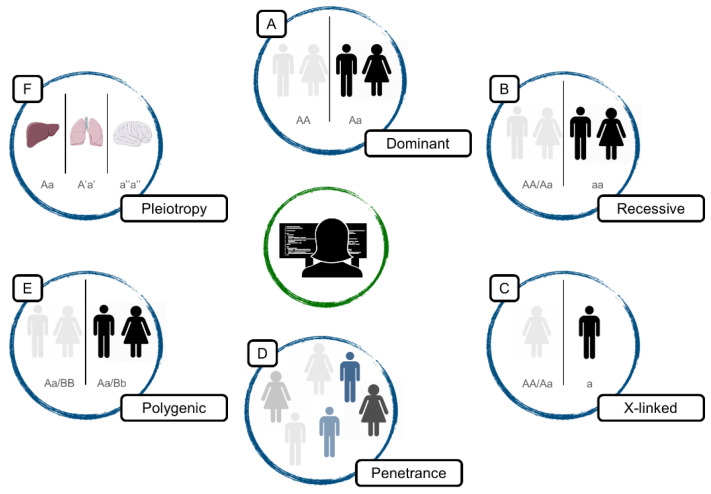
Examples of gene effect complexity. (**A**). Dominant: a single mutated allele is enough to cause the disorder (**B**). Recessive: two mutated alleles are required to cause the disorder (**C**). X-linked: males that carries the disease-causing mutation are affected due to the single copy of the X chromosome (**D**). Penetrance: same genetic variant might not develop the same symptomatology in different individuals (**E**). Polygenic: disorder caused by the combined action of more than one gene (**F**). Pleiotropy: mutations in a single gene affects two or more apparently unrelated disorders.

**Figure 2 ijms-23-07176-f002:**
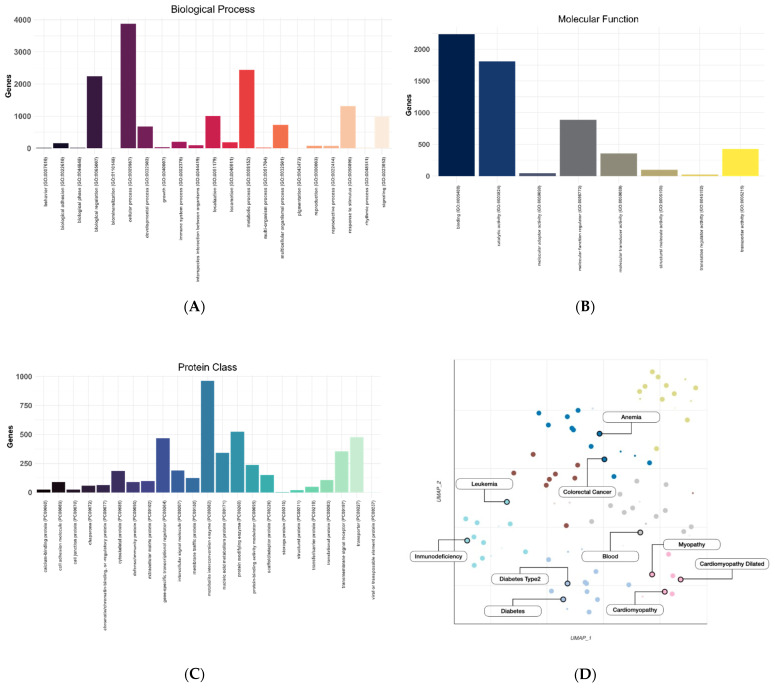
TSOex gene panel description: (**A**) Gene Ontology biological process (GO_BP) terms; (**B**) molecular function (GO_MF) terms; (**C**) protein class terms; (**D**) OMIM disease UMAP clustering. A, B, and C were determined using PANTHER GO-Slim gene lists. Fisher’s exact test was performed using an adjusted *p*-value < 0.05, calculated by the Benjamini–Hochberg method. Seven OMIM disease clusters were detected using EnrichR. From the 71 significantly enriched terms (q-value < 0.05; Appendix A), the 10 most significant are labelled in panel D (adjusted *p*-values were calculated using the Benjamini–Hochberg method).

**Figure 3 ijms-23-07176-f003:**
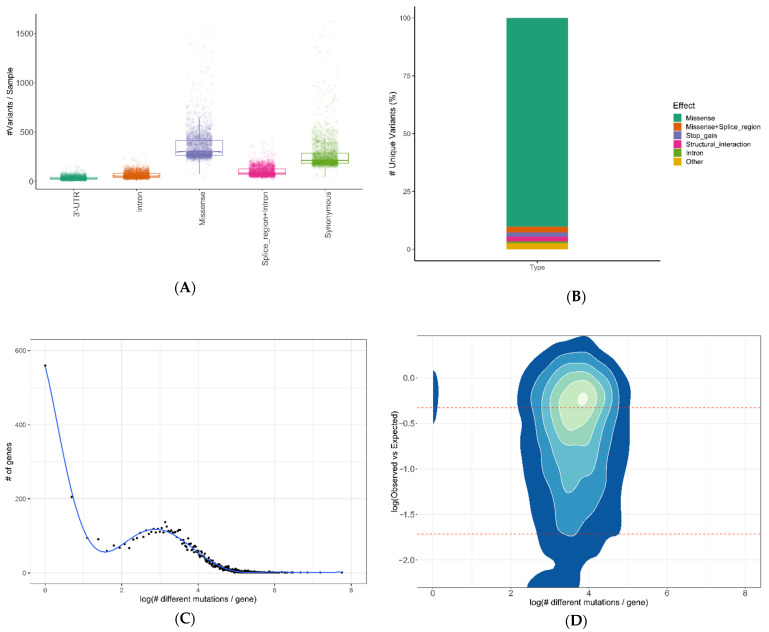
Genomic variability: (**A**) Number of variants detected per sample, classified by the five most abundant mutation classes. (**B**) Percentage of unique mutations detected in all 2474 patients, by class. (**C**) Genes sorted by the number of unique mutations detected (log-scaled). (**D**) Density map of gene constraints determined by gnomAD by the number of unique mutations detected (log-scaled). The red lines represent the first and forth quartiles for C and D. 3′UTR (variant in 3′ untranslated Region); intron (variant in non-coding region); missense (non-synonymous variant in coding region); splice_region + intron (splice-site variant in non-coding region); synonymous (variant in coding region that produces the same amino acid); stop-gain (variant that causes a stop codon); structural interaction (interaction loci that are likely to be supporting the protein structure).

**Figure 4 ijms-23-07176-f004:**
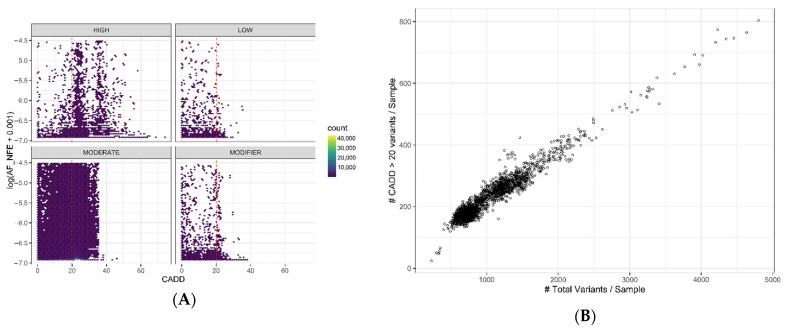
Variant conservation score: (**A**) Hexagonal heatmap of variant frequency related to the CADD score, grouped by the SnpEff impact classes (Appendix A). (**B**) Scatterplot of variants with high CADD scores (>20) related to total variants per sample. (**C**) Distribution of CADD scores of detected causative variants (n = 807). The red lines represent a CADD score of 20. AF_NFE stands for allele frequency_non-Finnish European population (gnomAD).

**Figure 5 ijms-23-07176-f005:**
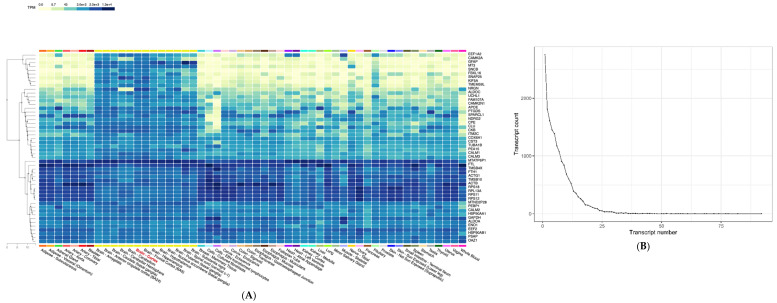
Transcription variability: (**A**) Tissue expression profiles of the 50 most expressed genes in the brain cortex from GTEx. (**B**) Number of protein-coding transcripts per gene using GTEx tissue data. (**C**) Rank 1 transcript switch in 1, 2, 3, or 4 out of 54 tissue datasets, using the TREGT database. Expression units are transcripts per million (TPM).

**Figure 6 ijms-23-07176-f006:**
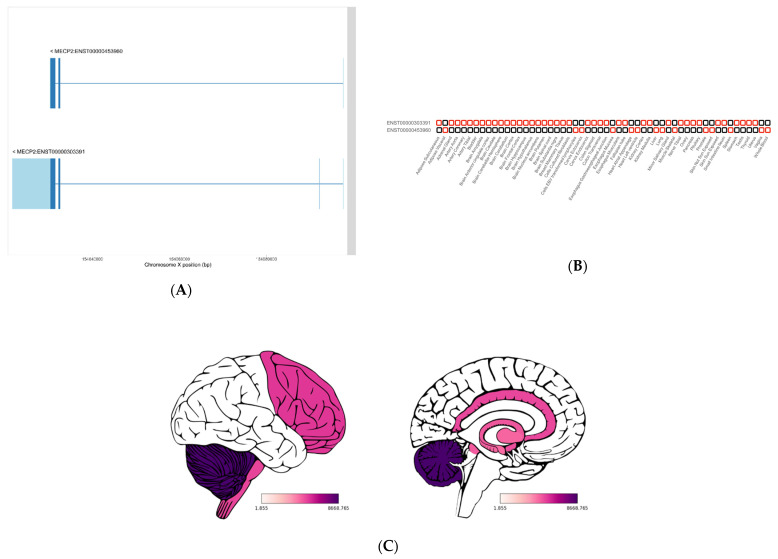
MECP2 transcript depiction: (**A**) MECP2 schematic diagram of the two major isoforms. (**B**) Transcript switch between the different MECP2 isoforms. (**C**) MECP2 gene expression in different brain areas. In A, UTR sequences are represented in light blue. In total, 14 rank 1 transcript switches are detected among the 54 tissues studied in the TREGT database. The GTEx brain expression profile (TPMs) is represented in C using the cerebroViz R package; cerebroViz output for exterior (left) and sagittal (right) views.

**Figure 7 ijms-23-07176-f007:**
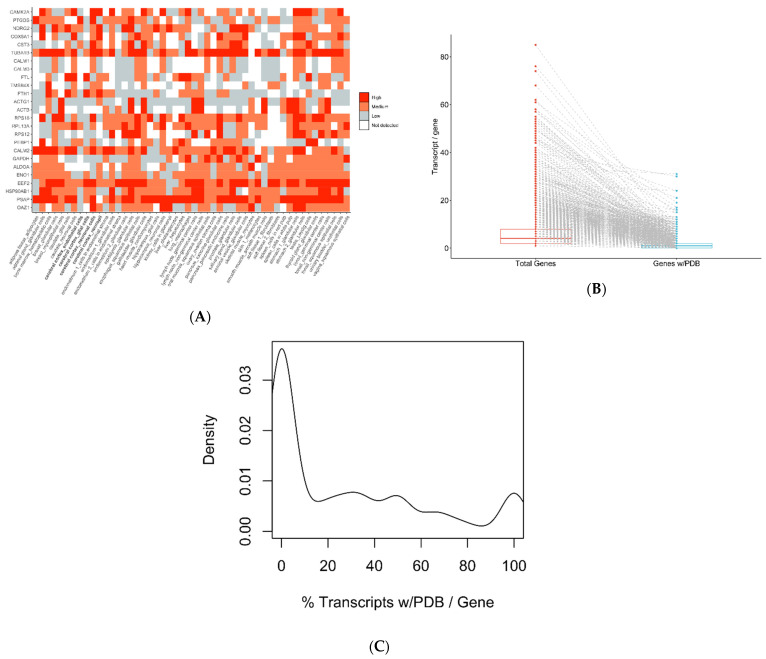
Protein detection and available PDB structures: (**A**) Tissue protein profiles of 25 of the 50 most expressed genes in the brain cortex from GTEx. (**B**) Gene transcripts from the TSOex panel with and without PDB IDs. (**C**) Transcript abundance with PDB IDs. In Figure 7A, proteins with no available data in >50% of the analyzed tissues from the Protein Cell Atlas were removed. In Figure 7B, the connecting dotted lines link the numbers of all transcripts from one gene with the amounts of these transcripts with PBD IDs.

**Figure 8 ijms-23-07176-f008:**
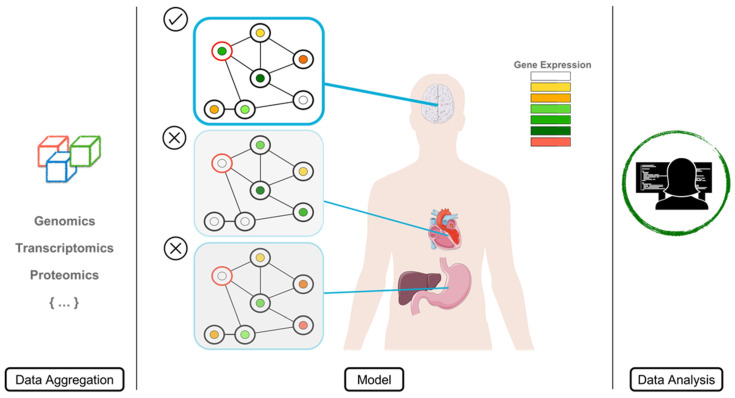
Multi-omics integrative analysis model.

**Table 1 ijms-23-07176-t001:** Variants conservation score according to its impact class.

IMPACT	Total Number	CADD > 20	%CADD > 20
HIGH	36,753	32,207	87.63
LOW	7010	800	11.41
MODERATE	884,264	513,406	58.06
MODIFIER	13,027	1536	11.79
	941,054	547,949	58.23

**Table 2 ijms-23-07176-t002:** Variant classification in one random sample.

IMPACT	Pathogenic	Likely Pathogenic	VUS ^1^	Likely Benign	Benign
HIGH	11	11	65	14	119
LOW	-	4	118	388	1284
MODERATE	-	7	186	284	722
MODIFIER	-	-	1368	476	3842
	11	22	1737	1162	5967

^1^: Variant of uncertain significance.

## Data Availability

Not applicable.

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
