# Peer review of "Molecular Modelling Hurdle in the Next-Generation Sequencing Era"

_ijms, 2022, doi:10.3390/ijms23137176_

Round 1
Reviewer 1 Report
The authors present the great difficulty that analyzes with NGS entail, namely the large number of variants of uncertain significance that emerge. They offer a complete overview of the tools available today, with limitations and advantages, and suggestions for integrating them in order to arrive at a diagnosis.
Minor advice:
Line 38: could the authors please insert a reference?
Line 98: remove the double space at the end of phrase
Line 138: I suggest to use "variants" insted of "mutations"
Line 196: remove the double dot
Tables: the table with the 71 omim genes is repeated in each file of tables and supplementary tables, I think it is a mistake
Author Response
Reviewer 1:
The authors present the great difficulty that analyzes with NGS entail, namely the large number of variants of uncertain significance that emerge. They offer a complete overview of the tools available today, with limitations and advantages, and suggestions for integrating them in order to arrive at a diagnosis.
Point#1. Minor advice:
- Line 38: could the authors please insert a reference?
Yes, we added the reference 2 on line 41.
- Line 98: remove the double space at the end of phrase.
Thank you, we removed it.
- Line 138: I suggest to use "variants" insted of "mutations".
We agree, we changed it on the text.
- Line 196: remove the double dot.
Thank you.
- Tables: the table with the 71 omim genes is repeated in each file of tables and supplementary tables, I think it is a mistake
We revised it and you are right, it was a mistake, we deleted all the extra-tables on each supplementary.
We used “Proof-Reading-Service.com” to correct our manuscript in order to ensure consistency of the spelling, grammar and punctuation, and to check the format of the sub-headings, bibliographical references, tables, figures etc. We have the certified if the reviewers or editor will need it.
We think that the revised version addresses all the major concerns of the referees. We thank you for your offer to review this paper, and hope that it will be acceptable for publication in International Journal of Molecular Sciences.
Yours faithfully,
Dèlia Yubero
Reviewer 2 Report
A very interesting premise in that variant of uncertain significance has now been more prominent as more WES and WGS are being conducted. Also in the abstract, the authors should take into consideration that it is not only the nonsynonymous mutations that are significant but the synonymous ones may also be important. The terms used across the manuscript need to be defined better.
The identification of one or multiple variants that trigger an RD maps a genomic localization to a function (genotype-phenotype). The obvious handicap of working with RDs is being able to establish a cause-effect association of a specific mutation. These two sentences are a bit confusing, please state whether the identification of the genetic alteration that you refer to as mapping in the first sentence is immediately linked to a phenotype, or does it need to go through cause and effect/ mechanistic tests?
In line 51, the second strategy may bear less accurate results than the first since the genetic alterations leading to the phenotype across a large population might be quite different (compared to strategy 1).
Moreover, due to pleiotropic effects, other factors 80 are putting molecular analysis alone into a deadlock: a single gene could affect multiple and apparently unrelated phenotypes and mutational penetrance, and a particular mutation does not always produce the same effect in all individuals who carry it. As the authors mention the picture is monumentally complex. I would suggest the authors could design a figure/ panel summarising the aspects that can lead to variability in the introduction.
In figure 1, please elaborate more on why metabolic genes, binding, and catalytic activity have been chosen. In addition, PANTHER uses various statistical tests, please specify which one was used for this GO analysis and what your p and q values were for sections A-C. Labels for section D are missing from the figure legend.
2A; missense mutations were the most prominent. Why?
Although synonymous mutations are also very represented, when determining the unique mutations within all samples, their proportion becomes almost negligible. What does this signify?
More than 90% of mutations are missense, followed by stop gain and structural interaction mutations. It would be good if the authors could provide definitions for missense, synonymous and nonsynonymous alterations in the intro to clearly outline what they exactly mean by each term.
In figure 2C, the authors could provide the full list of alterations in a supplementary file.
Due to the differential mutational landscape among the analyzed genes, we compared them to their intolerance of missense variation using the observed versus expected value from gnomAD (Figure 2D). Please define tolerance in this context. Also, provide examples for each category. Also in lines 152-159 please provide more interpretation.
In figure 3A, please provide a better narrative to explain the high-low categorisation, why the moderate category is more populated and the AFs observed in each category, and provide an example for each. Also, it is not very clear what high-low refers to (please explain SnpEff impact classes in more detail) and what this finding signifies.
When analyzing individual samples, there is a linear correlation (r = 0.97) between the number of total variants and the ones with a CADD score >20 (Figure 3B). Please specify the significance of this finding.
It is worth mentioning that a high concentration area can be observed in the MODERATE mutations, with a low allele frequency and a CADD score just beyond 20. Why has this happened?
From the 2,474 samples analyzed, approximately 800 (30%) had a single nucleotide variant (SNV) that could be associated with the patient’s pathophysiological process. SNVs represent more than 60% of selected variants responsible for a patient’s phenotype. According to our own data, diagnosis efficiency using TSOex reaches approximately 50%. Why has the association with pathology only improved by 30%?
variants can also be classified into five main groups, namely pathogenic, likely pathogenic, VUS, likely benign, and benign. Taking one sample as an example, variants were assigned to one of these five categories using the Varsome software. On what grounds does this software categorise these alterations.
As expected, most of this VUS belong to the MODIFIER category as its relevance in gene function or regulation is not well established. Please explain the snppef categorisation better.
Figure 4 is a bit off-topic since it describes 50 most expressed genes in the brain. Possibly the authors could show expression profiles of the actual genes used in this analysis, also with figures 5 and 6 please provide examples relevant to the original gene list. If they are relevant to your gene list please made this more evident. Also if the authors could provide global data for expression, protein variability, and structure relevant to their TSOex approach that would be great. Also linking the alterations with these profiles would also be really useful.
In general, the study started off strong but needs to better link the genetic alterations with the readouts in figures 4-6.
The texts in the figures are really small and difficult to read.
Author Response
Reviewer 2:
Point#1. A very interesting premise in that variant of uncertain significance has now been more prominent as more WES and WGS are being conducted. Also in the abstract, the authors should take into consideration that it is not only the nonsynonymous mutations that are significant but the synonymous ones may also be important. The terms used across the manuscript need to be defined better.
Thank you for your comment, we specified on the abstract the type of variation of unknown significance that is relevant for analysis. Regarding this comment and the ones that follow, we revised undefined terms over the manuscript.
Point#2. “The identification of one or multiple variants that trigger an RD maps a genomic localization to a function (genotype-phenotype). The obvious handicap of working with RDs is being able to establish a cause-effect association of a specific mutation.” These two sentences are a bit confusing, please state whether the identification of the genetic alteration that you refer to as mapping in the first sentence is immediately linked to a phenotype, or does it need to go through cause and effect/ mechanistic tests?
Thank you for the comment. In order to establish the causality of a certain variant to a phenotype, generally functional studies must be performed. We completed the sentence in order to give more comprehension into this part.
Point#3. In line 51, the second strategy may bear less accurate results than the first since the genetic alterations leading to the phenotype across a large population might be quite different (compared to strategy 1).
We agree, and we added a sentence regarding this point on line 56.
Point#4. “Moreover, due to pleiotropic effects, other factors are putting molecular analysis alone into a deadlock: a single gene could affect multiple and apparently unrelated phenotypes and mutational penetrance, and a particular mutation does not always produce the same effect in all individuals who carry it.” As the authors mention the picture is monumentally complex. I would suggest the authors could design a figure/ panel summarising the aspects that can lead to variability in the introduction.
We agree that it would be interesting to show this complexity, so we have added a new figure illustrating this, trying to present diverse concepts that add complexity to the variant analysis. The new figure (figure 1) is referenced on line 86.
Point#5. In figure 1, please elaborate more on why metabolic genes, binding, and catalytic activity have been chosen. In addition, PANTHER uses various statistical tests, please specify which one was used for this GO analysis and what your p and q values were for sections A-C. Labels for section D are missing from the figure legend.
The biological processes shown in figure 1 (now figure 2) were not selected from a list, but enriched in TSOexpanded panel, so these terms define the target panel we are using, which contains an invariable list of genes. We added Panther’s statistical test on figure 1 legend. Moreover, we specified that labels were added in panel D.
Point#6. 2A; missense mutations were the most prominent. Why?
In the TSO expanded data, we are capturing the coding regions from a restricted set of genes and approximately 20 bp of intronic regions, thus we expect that variants in coding regions are more common within this panel. Moreover, we have selected the variants with global allele frequency <2%, and possibly synonymous variants tend to be more polymorphic than the non-synonymous and are removed from the analysis. Therefore, the commercial panel biases these results and the filtered data selected to show as relevant variants that we believe are potentially significant for phenotype.
Point#7. Although synonymous mutations are also very represented, when determining the unique mutations within all samples, their proportion becomes almost negligible. What does this signify?
As we mention in the text, the drastic reduction in recurrent synonymous mutations might be the result of common variants in the Southern European population that are underrepresented in the gnomAD database.
Point#8. More than 90% of mutations are missense, followed by stop gain and structural interaction mutations. It would be good if the authors could provide definitions for missense, synonymous and nonsynonymous alterations in the intro to clearly outline what they exactly mean by each term.
You are right, we have explained the definition of these terms on Figure 2 legend that is where these terms firstly appear, as it does not fit perfectly on the introduction. We also explained other terms, as suggested previously.
Point#9. In figure 2C, the authors could provide the full list of alterations in a supplementary file.
Figure 2C (now figure 3) refers to the amount of mutations per gene. The total number of alterations we are considering with our data is of 941054, as indicated in Table 1. We do not consider this information to be relevant for the aim of the paper.
Point#10. Due to the differential mutational landscape among the analyzed genes, we compared them to their intolerance of missense variation using the observed versus expected value from gnomAD (Figure 2D). Please define tolerance in this context. Also, provide examples for each category. Also in lines 152-159 please provide more interpretation.
We substituted intolerance on line 159 for the meaning we pretended to explain. We added gene examples for the categories that are mentioned in the text.
Point#11. In figure 3A, please provide a better narrative to explain the high-low categorisation, why the moderate category is more populated and the AFs observed in each category, and provide an example for each. Also, it is not very clear what high-low refers to (please explain SnpEff impact classes in more detail) and what this finding signifies.
SnpEff definition of high, moderate, low and modifier is provided on supplementary table 2, but we considered your comment and also added the definitions in the main text (lines 210-217). Also, we provided examples for each SnpEff category, and added them to the supplementary table S2.
Point#12. When analyzing individual samples, there is a linear correlation (r = 0.97) between the number of total variants and the ones with a CADD score >20 (Figure 3B). Please specify the significance of this finding.
We have added the sentence “Conservation scores, although relevant, most pathogenic variants fall into the high CADD scores (Fig. 3C) (now figure 4) we show to be less specific than desired with a 20% total detected variants with high values (Fig. 3B).” on line 439 in order to explain better this finding.
Point#13. It is worth mentioning that a high concentration area can be observed in the MODERATE mutations, with a low allele frequency and a CADD score just beyond 20. Why has this happened?
In the moderate variants we find the higher proportion of VUS, low frequency variants with one or two ítems indicating pathogenicity, but insufficient evidence to classify as pathogenic, so further studies like segregation with parent samples and correlation with phenotype are required. It is expected that these are the most common in our panel because it is a targeted design and we have index patients, probably performing approximations like WES-trio the high number of moderate variants with high CADD scores would reduce this number. We have added a sentence on line 245 regarding this comment.
Point#14. From the 2,474 samples analyzed, approximately 800 (30%) had a single nucleotide variant (SNV) that could be associated with the patient’s pathophysiological process. SNVs represent more than 60% of selected variants responsible for a patient’s phenotype. According to our own data, diagnosis efficiency using TSOex reaches approximately 50%. Why has the association with pathology only improved by 30%?
We amended the sentence to better clarify this point, we believe that it was not well explained.
Point#15. variants can also be classified into five main groups, namely pathogenic, likely pathogenic, VUS, likely benign, and benign. Taking one sample as an example, variants were assigned to one of these five categories using the Varsome software. On what grounds does this software categorise these alterations.
The Varsome software classifies variants following the ACMG-AMP guidelines, already referenced on the manuscript (reference 11), and have added a short sentence on line 249.
Point#16. As expected, most of this VUS belong to the MODIFIER category as its relevance in gene function or regulation is not well established. Please explain the snppef categorisation better.
We agree, this issue has already been answered and fixed on the main text as you previously suggested.
Point#17. Figure 4 is a bit off-topic since it describes 50 most expressed genes in the brain. Possibly the authors could show expression profiles of the actual genes used in this analysis, also with figures 5 and 6 please provide examples relevant to the original gene list. If they are relevant to your gene list please made this more evident. Also if the authors could provide global data for expression, protein variability, and structure relevant to their TSOex approach that would be great. Also linking the alterations with these profiles would also be really useful.
We want to clarify that we have not acquired data from transcriptomics and proteomics from our patients, our intention was to expose a model that allows exploiting these kind of data in order to improve genetic analysis from DNAseq data. In fact, from the 50 most expressed genes in the brain that appear on figure 4A, only 23 are included in the TSOex panel. Anyway, we decided to show all of them because we expect that our model is relevant for broader NGS data like WES or WGS. This also explains for figure 6. Regarding figure 5, the MECP2 gene was selected in order to show an example, and this gene is included in the TSOex gene list. In addition, on the materials and methods section 4.5 we specify that transcriptomics and protein data are obtained from different databases (GTEx, TREGT, Human Protein Atlas).
Point#18. In general, the study started off strong but needs to better link the genetic alterations with the readouts in figures 4-6.
Thank you for the comment.
Point#19. The texts in the figures are really small and difficult to read.
We will offer editor figures in high resolution to increase the text font.
We used “Proof-Reading-Service.com” to correct our manuscript in order to ensure consistency of the spelling, grammar and punctuation, and to check the format of the sub-headings, bibliographical references, tables, figures etc. We have the certified if the reviewers or editor will need it.
We think that the revised version addresses all the major concerns of the referees. We thank you for your offer to review this paper, and hope that it will be acceptable for publication in International Journal of Molecular Sciences.
Yours faithfully,
Dèlia Yubero

Round 2
Reviewer 2 Report
The authors have addressed my comments.